# Evolution of the SARS-CoV-2 Omicron Variants: Genetic Impact on Viral Fitness

**DOI:** 10.3390/v16020184

**Published:** 2024-01-25

**Authors:** Wenhao Liu, Zehong Huang, Jin Xiao, Yangtao Wu, Ningshao Xia, Quan Yuan

**Affiliations:** 1State Key Laboratory of Vaccines for Infectious Diseases, Xiang An Biomedicine Laboratory, School of Public Health, Xiamen University, Xiamen 361000, China; liu18002000@163.com (W.L.); nsxia@xmu.edu.cn (N.X.); 2National Institute of Diagnostics and Vaccine Development in Infectious Diseases, Xiamen University, Xiamen 361102, China

**Keywords:** SARS-CoV-2, Omicron, spike, fitness, immune evasion, cell infectivity, cross-species

## Abstract

Over the last three years, the pandemic of COVID-19 has had a significant impact on people’s lives and the global economy. The incessant emergence of variant strains has compounded the challenges associated with the management of COVID-19. As the predominant variant from late 2021 to the present, Omicron and its sublineages, through continuous evolution, have demonstrated iterative viral fitness. The comprehensive elucidation of the biological implications that catalyzed this evolution remains incomplete. In accordance with extant research evidence, we provide a comprehensive review of subvariants of Omicron, delineating alterations in immune evasion, cellular infectivity, and the cross-species transmission potential. This review seeks to clarify the underpinnings of biology within the evolution of SARS-CoV-2, thereby providing a foundation for strategic considerations in the post-pandemic era of COVID-19.

## 1. Molecular Biological Characteristics of SARS-CoV-2

Coronavirus disease 2019 (COVID-19) is caused by severe acute respiratory syndrome coronavirus 2 (SARS-CoV-2), a positive-sense, single-stranded enveloped RNA belonging to the Coronaviridae family [1]. Consistent with SARS-CoV and MERS-CoV, SARS-CoV-2 also belongs to the betacoronavirus and has a genomic structure similar to other betacoronaviruses [2]. The organization of the virus genome is 5′-UTR-replicase (ORF1a/ORF1b)-S (spike)-E (envelope)-M (membrane)-N (nucleocapsid)-3′ UTR-poly (A) tail with accessory genes interspersed within the structural genes [3] (Figure 1). It contains a 5′ cap structure along with a 3′ poly (A) tail, which enables it to function as messenger RNA (mRNA) for translating the replicase polyproteins. The replicase gene, which encodes the non-structural proteins (NSPs), occupies approximately two-thirds of the genome, about 20 kb. In contrast, the structural and accessory proteins comprise only about 10 kb of the viral genome. The 5′ end of the genome includes a leader sequence and untranslated region (UTR) featuring multiple stem-loop structures that are essential for RNA replication and transcription. Furthermore, transcriptional regulatory sequences are present at the beginning of every structural or accessory gene, facilitating the expression of these genes. The 3′ UTR also harbors RNA structures necessary for viral RNA synthesis and replication. Four structural proteins are essential for virion assembly and infection of the virus. The M protein has three transmembrane domains and it shapes the virions, promotes membrane curvature, and binds to the nucleocapsid [4,5]. The E protein contributes to virus assembly and release and is involved in viral pathogenesis [6,7]. The N protein facilitates the encapsulation of viral genome RNA into a long helical ribonucleocapsid (RNP) complex. It actively engages in virion assembly by interacting with both the viral genome and the membrane protein M [8,9]. Studies have reported that the N protein can also interact with NSP3 protein, which assists in anchoring the genome to the replication–transcription complex (RTC) and facilitating the packaging of the encapsidated genome into virions. This interaction between the N protein and NSP3 is also crucial for the viral replication process [3,10,11]. Interspersed between these structural proteins are nine accessory proteins: ORF3a, ORF6, ORF7a, ORF7b, ORF8, ORF9b, ORF10 and ORF14. These accessory proteins exhibit considerable variability across different coronaviruses, yet they maintain a certain degree of conservation within each viral species. Although not directly involved in viral replication, these proteins play critical roles in facilitating the virus’s evasion of the host immune response [12].

The surface-anchored viral glycoprotein S protein facilitates virus entry into host cells. It accomplishes this by initiating a two-step process: first binding to a primary host-cell surface receptor, then catalyzing the fusion of viral and host-cell membranes [13]. The S protein serves as a critical regulator of viral attachment due to its ability to penetrate the host-cell membrane. Furthermore, it significantly contributes to defining the virus’s virulence, determining the target tissues, and the diversity of the host [14]. As a member of the class I viral membrane fusion proteins, the S protein consists of three segments: an ectodomain, a single-pass transmembrane anchor, and a short intracellular tail [15]. The ectodomain can be divided into a receptor-binding S1 subunit and a membrane-fusion S2 subunit. S1 contains two independent domains: an N-terminal domain (NTD) and a C-terminal domain (CTD) [16]. The receptor-binding domain (RBD) on S1 can specifically recognize angiotensin-converting enzyme 2 (ACE2) as its receptor [17]. When RBD binds to the host receptor ACE2, the conformational of the S2 subunit changes and the cleavage site on S2 is exposed and cleaved by host proteases, which allows the fusion peptide to insert itself into the host target cell membrane. The heptad repeat 1 (HR1) region in the S2 subunit forms a homotrimeric assembly, which exposes three highly conserved hydrophobic grooves on the surface that bind heptad repeat 2 (HR2). This six-helix bundle (6-HB) core structure is formed during the fusion process and helps bring the viral and cellular membranes into proximity for viral fusion and entry [18]. Identified as a typical genomic difference from SARS-CoV, four amino acid (aa) residues (PRRA) have been inserted at the junction of S1 and S2 subunits in SARS-CoV-2. This insertion generates a polybasic cleavage site (RRAR), which allows for effective cleavage by furin and other proteases [19]. Structural studies indicate that the furin cleavage site can reduce the stability of the SARS-CoV-2 S protein and facilitate the conformational adaptation needed for the binding of the RBD to ACE2 [20].

## 2. The Evolution of SARS-CoV-2 Variants

The first indication of SARS-CoV-2 genetic evolutionary selection pressure became evident as a variant (such as B.1) containing the S protein mutation D614G emerged in Europe in early 2020 and rose to a prevalence of nearly 100% by June 2020 [21,22]. Then, several variants of concern (VOCs) emerged, including Alpha (known as B.1.1.7 with PANGO nomenclature) first reported in England, Beta (alias of B.1.351) in South Africa, Gamma (alias of P.1) in Brazil, Delta (alias of B.1.617.2) in India and Omicron (alias of B.1.1.529) in southern Africa. It was reported that these variants are associated with an increase in the transmission or mortality of COVID-19 [23,24,25,26,27,28] or could escape immunity induced by the wild-type (WT) strain or D614G variant [29] (Figure 2A). The figure reveals a linear increase in the number of amino acid substitutions over time, with nearly all substitution counts (either of all genes or S protein) of significant variants aligning closely with the regression line. The average number of amino acid substitutions in all genes is approximately 25, while in the S protein, it is around 15. This observation indicates that the evolution of SARS-CoV-2 may adhere to specific patterns that need further examination and validation through rigorous statistical analysis (Figure 2B).

On 11 November 2021, B.1.1.529 was initially detected in Botswana, it rapidly caused the 4th wave of the SARS-CoV-2 epidemic in southern African countries, and swept the world in just a few weeks, causing a dramatic rise in infection numbers [30]. The epidemiological data indicated that the average basic reproduction number (R0), the estimate of the contagiousness when there is zero immunity or restrictive measure in the population, of the WT and Delta variants are 3.28 (95% CI: 1.4–6.5) and 5.08 (95% CI: 3.2–8.0) respectively [31,32]. For the Omicron variant, the average R0 range from 9.5 to 10.0 (95% CI: 5.5–24.0), meaning that each infected person, on average, infected 9.5 to 10 people. The average effective reproduction number (R_e_), the estimate of the contagiousness effected by immunity or restrictive measures in the population, for Omicron is 3.4–4.2, which is 2.7–3.8 times higher than that for the Delta variant (R_e_ of Delta is about 1.25–1.57) [33,34,35]. Given the severe threat posed by this variant to global health, the World Health Organization (WHO) designated it as a VOC and named it Omicron on 26 November 2021. The origin of the Omicron variant remains incompletely understood. The Omicron sublineages BA.1 (alias of B.1.1.529.1), BA.2 (alias of B.1.1.529.2), and BA.3 (alias of B.1.1.529.3) were discovered nearly around the same time as two of them (BA.1 and BA.2) have been prevalent globally, which may suggest Omicron had a long time to diversify before it was discovered [36]. BA.1 is characterized by a large number of changes in its S protein (Figure 2C), including 30 aa substitutions, three deletions, and an insertion, compared with the ancestral SARS-CoV-2. Based on changes in virological characteristics, including more infective and immunity-evasive, BA.1 rapidly supplanted previous variants, inclusive of the Delta, whereas many people have already been vaccinated or recovered from a previous infection. Then BA.2 further replaced BA.1 as the predominant circulating SARS-CoV-2 variant, demonstrating a robust capacity for antibody evasion and higher transmissibility [37].

Since the emergence of the BA.2 sublineage, it has become the dominant strain globally, with approximately 4.2 and 1.5 times greater infectivity than Delta and BA.1 [38]. Then the BA.2 sublineages BA.4, BA.5, and BA.2.75, gradually replaced their paternal strain, causing a new surge across the world. The estimated R_e_ for BA.4 and BA.5 in the United States are 1.28- and 1.36-fold higher than that of BA.2, respectively [39]. From a molecular point of view, the S mutations L452R, F486V, and R493Q of BA.4 and BA.5 are located in the RBD (Figure 2C), promoting both the viral attachment to the human cells and immune evasion [40]. Remarkably, F486V results in immune evasion and decreased binding affinity to ACE2, whereas L452R compensates for the decreased ACE2 binding affinity. This may be related to one of the SARS-CoV-2 evolution strategies—the balance between immune evasion and ACE2 affinity [39]. On the other hand, there are 9 additional mutations G339H, G446S, G257S, I210V, F157L, R493Q, N460K, W152R, and K147E in the S protein of BA.2.75 compared to BA.2 [41], which may bring BA.2.75 greater membrane fusogenic, intrinsic pathogenicity, and humoral evasion than BA.2 or BA.5. The R_e_ of BA.2.75 are 1.34- and 1.13-fold higher than that of BA.2 and BA.5, respectively [42]. The BA.5 sublineage BQ.1.1, harbors the mutations K444T, N460K, and R346T in critically antigenic sites of S protein (Figure 2C). Owing to its advantage in immune escape over other circulating Omicron subvariants, BQ.1.1 demonstrates a notable transmission advantage, consequently supplanting the previously dominant BA.5 in many countries [43]. When the R_e_ of BA.5 is set at 1, the relative R_e_ values for BQ.1 and BQ.1.1 are 1.23 and 1.24 [44,45].

The XBB, the recombinant of BJ.1 (alias of B.1.1.529.2.10.1.1) and BM.1.1.1 (alias of B.1.1.529.2.75.3.1.1.1), was first reported on 19 August 2022 and has emerged as the dominant subvariant starting from April 2023. Comparative analysis revealed that XBB and XBB.1 exhibit R_e_ that are 1.24- and 1.26-fold higher than BA.5 and are comparable with those of BQ.1 and BQ.1.1 [44]. Strikingly, the S protein of the predominant XBB subvariant has 14 mutations in addition to those found in BA.2, including 5 in the NTD and 9 in the RBD. Following the convergent acquisition of the F486P mutation, which is related to the RBD-ACE2 binding affinity [46], the XBB sublineages propagated expeditiously on a global scale. Among the various sublineages, XBB.1.5, XBB.1.9.1, XBB.1.9.2, and XBB.1.16 have emerged as the predominantly prevalent strains. These subvariants exhibit similar antigenic characteristics associated with the S protein. Like F486P, the F456L mutation in the S protein is also prevalently found in the offspring of the previous dominant XBB sublineages, due to an enhancement in fitness. A series of new convergent evolutionary strains, represented by EG.5.1, FL.1.5.1, and XBB.1.16.6, gradually began to dominate in the latter half of 2023 [47].

On 17 August 2023, the World Health Organization (WHO) designated BA.2.86 as a variant under monitoring (VUM) due to the significant number of mutations observed, which was one of the highest since the emergence of Omicron. BA.2.86 was considered a derivative of BA.2 and possesses 33 S protein mutations and 14 RBD mutations compared with BA.2 and 35 S protein mutations and 12 RBD mutations compared with XBB.1.5 [48]. In addition to the mutations common with XBB.1.5, BA.2.86 RBD harbors extra mutations— I332V, K356T, V445H, N450D, N481K, A484K, and 483del (Figure 2C)—which are reported to likely augment its capacity for immune evasion. Moreover, unusual mutations in NTD might alter the antigenicity of BA.2.86 [48]. These findings imply that BA.2.86 may possess higher transmissibility than the currently prevalent XBB variants, including EG.5.1.

## 3. Hypotheses about the Origin of Omicron

Phylogenetic analysis of global SARS-CoV-2 sequences has not revealed any close intermediary sequences between Omicron and its closest relatives, therefore the origin of Omicron remains unclear [49]. The evolutionary analysis did not reveal any special mutational profile or frameshift event that could suggest that it descends from the Alpha, Beta, Delta, or Gamma variants. The very long branch of the Omicron lineage in a time-calibrated tree might reflect a cryptic and potentially complex evolutionary history [50]. Most mutations in Omicron were rarely reported among previous variants, supporting three prevalent hypotheses regarding its evolutionary history [51].

The most popular hypothesis regarding the proximal origins of Omicron suggests that Omicron may have emerged in a population with prolonged viral replication, possibly in an immunocompromised individual that provides a suitable host environment conducive to long-term intra-host virus adaptation. A delicate balance has been achieved between host adaptability immunity and viral escape, which allowed the virus to gradually accumulate a significant number of mutations over a considerable length of time, thereby leading to the distinct genetic makeup currently observed in the Omicron.

The second hypothesis proposes that Omicron could have accumulated mutations in a nonhuman host and then leaked into humans. A study has indicated that the range of mutations acquired by the Omicron progenitor notably differed from those of viruses that evolved in human patients. However, it showed similarity with the mutation patterns typically associated with viral evolution in a mouse cellular environment. Moreover, another study indicated that an earlier variant of SARS-CoV-2 could have acquired mutations that increased its potential to infect rodents from an ill person likely through contaminated sewage leading to its evolution into Omicron in the rodent population [52].

The third hypothesis proposes that the ancestor of the Omicron may have been prevalent in regions with under-developed SARS-CoV-2 genomic surveillance for a significant duration. The evolution of its intermediate-generation virus may have gone undetected due to inadequate surveillance capabilities [51,53].

As these hypotheses are speculative, additional research is necessary to ascertain the authentic evolutionary history of the Omicron. Developing an understanding of the origins and mechanisms that underlie its unique genetic composition will not only provide crucial insights for public health efforts but also offer a deeper comprehension and understanding of the virus-host interaction mechanisms and the systematic features of viral evolution.

## 4. Impacts of Mutations in Omicron on Viral Fitness

The field of viral fitness was originally developed through studies of a relatively small number of bacteriophage, animal, and plant viruses. With increasing recognition of the importance of viral fitness, there is now a wide array of study systems, and the majority of them are based on RNA viruses, and the highest number of publications in recent years involve human pathogens associated with major disease emergence events, such as human immunodeficiency virus-1 (HIV-1), influenza virus, and dengue virus (DENV) [54]. The broad definition of fitness is reproductive success, which is heavily influenced by the environment and often varies across time and location. The fitness of a given SARS-CoV-2 variant relies on changes in the immune characteristics of the host population, the particular traits of the variant’s competition within the viral population, and the stochastic sampling process. Immune evasion increased cellular infectivity, and even species tropism are crucial factors that contribute to viruses’ fitness advantage. The impact of biological factors on fitness dynamically changes. During the early stages of a viral outbreak, due to the lack of sufficient immunological pressure, mutations of immune evasion do not become the dominant factor driving the evolution of the virus. Instead, the enhancement of cellular infectivity and the adaptation to a broader range of host species may confer greater benefits in terms of fitness [55]. Meanwhile, as the scope of viral spread expands and the coverage of vaccinations increases, the impact of immune evasion on viral fitness will progressively intensify [56]. The evolution of SARS-CoV-2 after Omicron represents a new typical phase that means a change in the biological function and antigenicity of SARS-CoV-2. Here we primarily focus on how the mutations in Omicron and its subvariants influence fitness, and the elaboration will be conducted from three following aspects: immune evasion, cell infectivity, and cross-species tropism (Figure 3).

### 4.1. Immune Evasion

In response to the SARS-CoV-2 pandemic, a global race to develop vaccines has been in motion since 2020, leading to the rapid emergence of various types of vaccines, which offers the potential for constructing a global immunity barrier swiftly [56]. However, under the immense vaccination campaign and extensive transmission of SARS-CoV-2, the adaptive immunity obtained through either natural infection or vaccination has not been successful in establishing an infection-blocking immunity barrier among the population. This inability has gradually led to immune evasion, specifically humoral immune evasion, becoming a driving factor for the virus to enhance its adaptability for transmission among the population. There are various distinct mechanisms by which mutations can alter the antigenic properties of a glycoprotein, such as aa substitutions that alter the epitope, increasing receptor-binding avidity, changes in glycosylation, deletions, insertions, and allosteric structural effects. Alterations in the antigenicity of the S protein can induce modifications in virulence and host-pathogen dynamics, raising substantial concerns regarding immune escape following prior infections or vaccinations. The continual emergence of breakthrough infections and reinfections are notable characteristics of the Omicron phase [57].

Mutations in the S protein components are strongly associated with immune evasion by emerging variants. Studies have shown that mutations in the RBD of the S protein significantly reduce antibody binding, highlighting the importance of immunodominant epitopes within the RBD [58]. In fact, nearly half of the aa mutations in the Omicron variant are located in the RBD. Within the RBD, the receptor-binding motif (RBM, 438aa–508aa) harbors 12 mutations in the Omicron variant, with half of these mutations located around the N501 at the C-terminus. Notably, the mutation E484 in the RBM is a frequent site of mutation that enables escape from monoclonal antibodies, with aa changes to K, Q, or P resulting in a significant reduction in neutralization titers [59]. The Omicron variant carries the E484A mutation, while the Beta variant harbors the E484K mutation, both of which contribute to immune evasion [60]. Additionally, mutations in the NTD of the S protein have been observed to play a role in immune evasion. These mechanisms include insertions, acquisition of additional glycosylation motifs, and deletions in the NTD, which can lead to conformational changes and enhance the virus’s immune evasion capabilities [61]. Deletions in the NTD have been observed in the evolution of SARS-CoV-2 and are partly responsible for changes in its antigenicity [61,62,63]. Most studies on immune evasion in the NTD have focused on a region centered on a conformational epitope composed of residues 140–156 (N3 loop) and 246–260 (N5 loop) [64]. Other mutations, such as C136Y and S12P, which impact the epitope-paratope interface indirectly, have been shown to affect the neutralizing activity of several monoclonal antibodies, likely by disrupting the disulfide bond and dislodging the supersite targeted by these antibodies [62].

#### 4.1.1. Neutralization by Immune Sera

The emerging Omicron variant successively demonstrates a significant immune escape from polyclonal antibodies induced by vaccination or previous infection. Multiple subvariants can gain a fitness advantage through evasion of immune response in the human population. Since BA.1, the virus has exhibited a significant ability to escape from vaccines and the serum of recovered individuals. In a study, the neutralizing activity against both D614G and BA.1 pseudoviruses was examined using sera from individuals who were administered one of the four widely utilized COVID-19 vaccines, including BNT162b2 (Pfizer), mRNA-1273 (Moderna), Ad26.COV2.S (Johnson & Johnson), and ChAdOx1 nCoV-19 (AstraZeneca), were evaluated. Notably, a significant reduction in neutralizing potency was observed against BA.1 in all cases, and the two mRNA-based vaccines, BNT162b2 and mRNA-1273, even exhibited a >21-fold and >8.6-fold decline in titer, respectively [57]. Furthermore, BA.1 was assessed against a set of human sera from convalescent COVID-19 patients who had been infected with the WT strain. The findings revealed that the average neutralization titer of BA.1 displays approximately an 8.4-fold decrease compared to D614G. However, the neutralization sensitivities of other VOCs and variants of interest (VOIs), Lambda, Mu, Alpha, Beta, Gamma, and Delta, declined were only 1.7-, 4.5-, 1.2-, 2.8-, 1.6-, and 1.6-fold respectively, in comparison to D614G [58]. The successive emergence of other Omicron subvariants has revealed further escape capabilities. These capabilities also enable evasion of the antibodies induced by other Omicron sublineages. A study comprising the humoral immune response of five cohorts (3 shots WT, 4 shots WT, 3 shots WT + WT/BA.5 bivalent, BA.2 breakthrough, and BA.5 breakthrough) demonstrated that compared to D614G, the geometric mean titers against BA.2 and BA.4/5 decrease by 2.9- to 7.8-fold and 3.7- to 14-fold, respectively [59]. The subvariants BQ.1, BQ.1.1, XBB, and XBB.1 further exhibit a significant reduction in neutralization compared to their predecessors BA.5 and BA.2. In the cohorts of BA.2 and BA.4/5 breakthrough infections, the geometric mean titers against BQ.1, BQ.1.1, XBB, and XBB.1 show a reduction of 13- to 31-fold and even 86- to 135-fold, respectively [59]. A study involving cohorts of three doses of WT + WT/BA.5 bivalent vaccinees, BQ breakthrough infections, and XBB breakthrough infections found that the recently emerging subvariants EG.5 and EG.5.1 display significantly greater resistance (1.7-fold) to serum nAbs compared to the previously dominant subvariant XBB.1.5. This increased resistance is primarily attributed to the F456L mutation in the S protein that impacts antibody binding to the class I region of the RBD [60]. A study of three cohorts (3 shots WT, 2 shots WT + WT/BA.5 bivalent, and XBB.1.5 breakthrough) revealed varying degrees of resistance to serum neutralizing antibodies (nAbs) against BA.2.86 in comparison to XBB.1.5 and EG.5.1. Additionally, BA.2.86 displays a notable escape from nAbs in the sera of individuals who received a 3-dose monovalent vaccination, with nAb titers reaching or falling below the limit of detection. A neutralization assay conducted with XBB breakthrough infection sera revealed that the neutralization titer against BA.2.86 is lower (1.6-fold) compared to that against EG.5.1 [61]. However, the difference in resistance to serum nAbs from individuals with XBB-derived subvariant infection between BA.2.86 and XBB.1.5 is slight [62]. Fortunately, a recent report based on community sera from the UK suggested that the individuals received XBB.1.5-based vaccines confers broad-spectrum efficacy against BA.2.86, potentially enhances cross-protection against BA.2.86 and future lineages through stimulating the proliferation of pre-existing B cells [63]. Nonetheless, further research is necessary to determine if this cohort of healthy UK folks mentioned above is representative of the wider community, where hybrid immunity is prevalent.

#### 4.1.2. Neutralization by Monoclonal Antibodies

Owing to its remarkable safety profile and potent virus-neutralizing capacity, monoclonal antibodies (mAbs) targeting SARS-CoV-2 have been widely adopted in the clinical treatment of COVID-19. Most of the licensed mAbs, or those under development, neutralize the virus by targeting its S protein to block the interaction between SARS-CoV-2 and ACE2. Given that the majority of mAbs target the immunodominant epitopes on the S protein, mutant variants can evade these mAbs, affording them a fitness advantage within the human population. This trend has become increasingly apparent following the emergence of the Omicron.

Upon the emergence of BA.1 and BA.2, a significant escape capability was observed in the majority of mAbs at that time, resulting in a substantial reconfiguration of the mAbs market. Regdanvimab, developed by Celltrion Inc. (Incheon, Republic of Korea), as well as the cocktail therapies Bamlanivimab/Etesevimab and Casirivimab/IImdevimab, developed by Eli Lilly and Regeneron respectively, have all forfeited their neutralization against BA.1 or BA.2. GSK’s Sotrovimab, and the cocktail therapies Amubarvimab/Romlusevimab (only approved in China, but both randomized controlled trials and real-world research robustly substantiate its clinical efficacy [64,65,66]) and Evusheld (Cilgavimab/Tixagevimab), developed by Brii Biosciences and AstraZeneca respectively, have also experienced varying degrees of decline in neutralizing activity, albeit still retaining some level of neutralization. In contrast, Eli Lilly’s Bebtelovimab, which gained approval after the appearance of Omicron on 11 February 2022, has demonstrated high-efficiency neutralizing activity against BA.1 and BA.2 [67,68,69]. Regrettably, later on 30 November 2022, the Emergency Use Authorization (EUA) for Bebtelovimab was officially revoked due to its insufficient efficacy against Omicron subvariants and it was no longer authorized for emergency use in the United States. Previous studies have indicated that the S371F, D405N, and R408S mutations found in the BA.2 and emerging Omicron variants may lead to substantial evasion of nAbs [70,71].

After the emergence of BA.4 and BA.5, the situation of viral escape became even more dire. Amubarvimab/Romlusevimab witnessed a loss in its viral neutralization capacity, and the neutralizing efficacy of Evusheld experienced a further decline. However, Bebtelovimab continued to retain its robust ability to neutralize the virus [68,72,73]. After BA.4 and BA.5, the virus exhibits a peak in mAbs escape. Due to subsequent mutations in the 444–446 epitope of the dominant viral strains, both Bebtelovimab and Evusheld lost their neutralizing activity against BQ.1.1, CH.1.1, and XBB. Additionally, the R346T, K444T, N460K in BQ.1.1, and the R346T, V455P, G446S, N460K, F486S and F490S in XBB and XBB.1 are contributed to the resistance to RBD mAbs. Furthermore, another study also showed that N450D, K356T, L452W, A484K, V483del, and V445H contributed to the enhanced mAbs evasion of BA.2.86 compared with XBB.1.5 [48]. Among the approved mAbs, only Sotrovimab retained partial neutralizing efficacy [56,59,74]. The mAbs with potent neutralizing activity against currently prevalent viral strains are in the developmental phase. Examples include S3H3 developed by Zhong Huang et al. [75] and SA55 developed by Yunlong Cao et al. [76], along with variants 12–16 and 12–19 developed by David D. Ho et al. [77]. Opting for conservative epitopes or employing spatial effects in a rational design approach may represent effective strategies against the incessant emergence of escape variants [77,78].

### 4.2. Cell Infectivity and Replication

Virus, host, and environmental factors influence whether a successful transmission occurs by governing the infectivity of the respiratory virus, the contagiousness of the infector, the susceptibility of the exposed individual, and the environmental stress on the virus [79]. Here we mainly discuss the virus-driven determinants. The propensity for respiratory viruses to be transmitted is affected by virus stability under environmental stress [80,81], which in turn is influenced by the composition and structure of the virus envelope [82,83], capsid [84], internal proteins and genomes [85] as well as the formation of viral aggregates [86].

#### 4.2.1. Stability and Binding Affinity to ACE2 of S Protein

Viral protein expression and modification may influence infectivity. Viral surface and internal proteins can impact transmissibility by determining the site of infection and interacting with specific host receptors that exhibit varying binding specificity and affinity [79]. Recent studies have revealed that Omicron sublineages have further improved binding affinities for ACE2 and stability in vitro, which correlates with their increasing viral transmission and infection [73,87]. It has been demonstrated that specific mutations within the BA.1 RBD modulate the affinity towards ACE2, with some enhancing it and others diminishing [87]. Consequently, the affinity of Omicron BA.1 RBD remains similar to that of the WT strain and Delta RBD, while the affinity of BA.2 RBD is 1.95-fold greater than that of BA.1, potentially due to the G496S mutation in BA.1 disrupting the polar interaction network between ACE2 D38 and RBD R498/Y449 [87,88] (Figure 4). Although the direct affinity between RBD and ACE2 did not increase, the BA.1 variant may enhance its binding affinity during the actual infection process by increasing conformational stability. The BA.1 Spike trimer, despite adopting an upright configuration, exhibits a significantly more compact architecture in the regions formed by three copies of S2 and RBD. When compared to the counterparts from the Delta variant, the increase in intersubunit interactions of BA.1 indicates a highly stabilized conformation. In the broader context of the Spike trimer, the aa changes N856K, N969K, and T547K result in the conversion of short-side chains to long-side chains. This facilitates the formation of three hydrogen bonds with D658, Q755, and S982 from neighboring subunits, pulling the three subunits closer together. Furthermore, the substitution of D796Y can stabilize the sugar at residue N709 from its adjacent subunit through the formation of a hydrogen bond. Interestingly, the new emerging subvariant BA.2.86 Spike trimer exhibits greater conformational stability than the parental BA.2 and XBB variants, particularly EG.5.1. This increased stability may be a result of the A570V mutation, which is thought to enhance hydrophobic interactions between protomers [89]. Consistent with the structural observations, thermal stability assays confirmed that the Omicron spike protein trimer is more stable than those of the wild-type and Delta variants, resulting in a higher affinity for ACE2 [90]. The N460K and D339H are critical substitutions that cause BA.2.75 RBD to have a significantly higher (10-fold) affinity to ACE2 compared to BA.2 [42] (Figure 4). The binding affinities of BA.4/5 RBD to ACE2 demonstrated a slight enhancement (1.5-fold) compared to BA.2 [59]. However, BQ.1, and BQ.1.1 do not significantly modify the binding affinity between the RBD and ACE2 when compared to their predecessor BA.4/5 [59], which indicates factors such as evasion rather than ACE2 affinity may be driving the evolution of these subvariants. In contrast, the RBD of XBB and XBB.1 subvariants exhibit slightly decreased binding affinity (2.1-fold) to ACE2 compared to their predecessor BA.2, which may be related to the mutations F486S and R493Q [59]. The binding affinity between the RBD of XBB.1.5 and ACE2 was found to be comparable to that of BA.2.75 and significantly stronger than that of XBB.1 and BQ.1.1. Given that neither XBB.1 or XBB.1.5 demonstrates significant differences in their escape capabilities, an observed increase in affinity (5.5-fold than XBB.1 and 2.3-fold than BQ.1.1) could arguably be the primary determinant of the significant growth advantage seen in XBB.1.5. This could be attributed to the S486P mutation leading to a notable enhancement in the binding affinity to ACE2. Moreover, the results indicated that the EG.5.1 S RBD shows a significantly reduced binding affinity compared to XBB.1.16 (1.8-fold) while exhibiting a slight increase compared to XBB.1 (1.2-fold) [91]. BA.2.86 RBD exhibits higher ACE2 binding affinity than that of XBB.1.5 (3.4-fold) and EG.5 (4-fold), indicating that the low infectivity in vitro (293T-ACE2 cells) may be due to other factors, probably the dynamic change in RBD up–down transition or the efficiency of membrane fusion [48]. The V445H and R493Q mutations augment ACE2 binding affinity by establishing hydrogen bonds between the RBD of BA.2.86 and the ACE2 receptor, influencing membrane fusion and entry into diverse target cells [48,61]. Overall, although there is some fluctuation in the binding affinity of S protein to ACE2 during the virus evolution process, there is a general trend towards enhancement. Therefore, continued monitoring the affinity of the virus is important.

#### 4.2.2. Tissue Tropism and Cell Entry

One proposed scenario for the evolution of SARS-CoV-2 is through adaptation to human host factors. This includes more efficient utilization of the factors that can facilitate virus infection and replication, such as transmembrane protease serine 2 (TMPRSS2), cathepsin, ADAM10, and ADAM17 proteases. Compared to previous variants, Omicron is less capable of infecting cells of the lower respiratory tract (which have higher levels of TMPRSS2) [92], and prefers to infect cells of the upper respiratory tract, characterized by higher levels of cathepsin, which is necessary for the entry by endocytosis [93]. The attenuated replication capacity in the lung cells of Omicron may be related to the shift in its entry pathway preference, opting for TMPRSS2-independent endosomal fusion. This represents a significant alteration in the biological behavior of the virus, potentially leading to changes in its tissue tropism.

Entry of SARS-CoV-2 and related coronaviruses can proceed via two routes: in the presence of TMPRSS2, the S2 site is cut directly on the surface of the host cell and SARS-CoV-2 enters by fusion. On the other hand, if the infected cell expresses insufficient TMPRSS2 or the virus–ACE2 complex does not encounter TMPRSS2, SARS-CoV-2 will be internalized by endocytosis. At this point, with the acid pH present inside the endosome, the specific enzyme cathepsin becomes active and is allowed to cut the S2 site, with the subsequent fusion of the membranes and the release of the viral RNA in the host cell [92,94]. Pre-Omicron variants prefer entrance into the host cell by fusion (TMPRSS2 dependent route), while Omicron predominantly uses the second route to enter the host cell by endocytosis [94], which can lead to preferences in the selection of the tissues that become infected and a different cellular tropism [93]. This could be considered a sign of SARS-CoV-2 gradually adapting to the human host environment and transitioning towards non-systemic infections. In vitro, the results have shown that the replication of BA.1 and BA.2 was significantly attenuated in lower respiratory tissues compared to the previous variants [93,95,96,97]. However, later studies showed that BA.2.12.1 and BA.4/5 more efficiently replicate in lower respiratory tissues than BA.2 and BA.1, but it is still weaker than the pre-Omicron variants [39,98]. Subsequent strains such as XBB.1.5 and EG.5.1 have followed this trend by continuing to exhibit lower infectivity than BA.2 towards the lower respiratory tract tissues [91]. Regarding the recently discovered BA.2.86, it appears to demonstrate a relatively higher infectivity (1.9–2.8-fold) towards the lower respiratory tract cell (CaLu-3) compared to BA.1, BA.2, BA.4/5, XBB.1.5 and EG.5.1. However, its infectivity remains significantly lower than that of the early WT strains [89]. In contrast, in 293T-ACE2 cells, pseudovirus of BA.2.86 does not demonstrate a significant change in infectivity when compared to D614G, with only a 1.4-fold increase. However, it displays a 2.6-fold reduction in infectivity relative to BA.2. Moreover, its infectivity is 1.8 to 2.1 times lower than that of other Omicron subvariants, encompassing XBB.1.5 and EG.5.1 [89]. Another study using live virus indicates that there are no significant differences between BA.2.86 and XBB.1.5 in terms of replication and spread among cells [62]. Overall, it remains to be determined whether BA.2.86 will exhibit increased lung tropism and thus enhanced pathogenesis compared to previous Omicron subvariants.

In summary, Omicron accomplished a milestone evolution of SARS-CoV-2 in 2021 by adjusting its preferences for cellular entry and tissue tropism. This evolution led to the manifestation of non-systemic infections, enabling the virus to fully adapt to human hosts, and thus spread within human societies in a significantly more efficient manner [99]. From then on, coexistence with SARS-CoV-2 has, regrettably, become the only option for humanity at large.

### 4.3. The Impact of Mutations beyond S Protein

While the evolution of SARS-CoV-2 exhibits a relatively lower degree of variation in the M, E, and N proteins relative to the S protein, the mutations in M, E, and N have significant implications for the virus’s biological characteristics because of the crucial roles played by them in the cellular infection process [100,101,102].

The N protein has garnered significant attention due to the mutations S202R and R203M. Research findings indicate that these mutations can result in an over 50-fold increase in viral production, potentially linked to improvements in the packaging and delivery efficiency of viral mRNA [103]. Another study suggests that the mutations R203K and G204R in the N protein increase nucleocapsid phosphorylation and confer resistance to GSK-3 kinase inhibition, providing a molecular basis for increased viral replication [104].

The M and E proteins exhibit a relatively high degree of conservation [100]. However, mutations such as D3G, Q19E, and A63T on the M protein are prevalent across various Omicron subvariants. Notably, the D3G mutation in the NTD of the M protein may significantly impact the virus–host cell interaction [105]. The E protein, a pivotal component in virus release, exhibits a prevalent T9I mutation in Omicron. This mutation is likely to impair the ion selectivity in the 2-E channel and decrease its pH sensitivity. Consequently, it diminishes cellular death and lowers cytokine production [106]. Such alterations may afford the virus a degree of fitness advantage by circumventing immune responses.

In addition to the structural proteins, mutations in accessory proteins, which are highly prevalent in various VOCs, can alter their secondary structure and impinge on their biological functions [107,108]. These proteins are vital in evading the immune response and facilitating virus pathogenesis [109,110]. Current research suggests that the mutations in non-structural proteins of SARS-CoV-2 may enhance the virus’s evasion against the host’s non-specific immune response. However, there is a dearth of in-depth studies on the impact of mutations in non-structural proteins on the adaptability of the virus.

NSP6 plays a critical role in mediating contact between double-membrane vesicles (DMVs) and the endoplasmic reticulum (ER) membrane, as well as channeling lipids to viral replication organelles [111]. The Omicron variant exhibits a three-amino-acid deletion in ORF1a at positions L3674, S3675, and G3676, which corresponds to a deletion in NSP6 at amino acids 105 to 107. It is speculated that this mutation may facilitate evasion of the innate immune response, potentially by impeding the infected cells’ capacity to degrade viral components [112,113]. Furthermore, it was suggested that the 105, 106, and 107 deletions and I189V mutation in NSP6 impair the lipid droplet (LD) channeling function of the protein [111], which is related to the attenuated phenotype of Omicron [114]. These observations support and further extend the findings of a previous study showing that mutations in the 5′-UTR–nsp12 region, in which NSP6 resides, contribute to Omicron’s attenuation in K18-hACE2 mice [115].

Furthermore, ORF 7a, 7b, and 8 are known to impede type I interferon (IFN-I) signaling, crucial for the host’s initial defense against viral infections. ORF8, in particular, has been demonstrated to interact with the major histocompatibility complex (MHC) class I, thus hindering antigen-presenting cell (APC) function [109]. Additionally, ORF7a and ORF7b have been implicated in binding with CD14+ monocytes, reducing their antigen-presenting capabilities and inducing a surge in proinflammatory cytokines, including IL-6, IL-1β, IL-8, and TNF-α [116,117]. During the early stages of the COVID-19 pandemic, a SARS-CoV-2 accessory sequence with a substantial 382-nucleotide deletion (Δ382), resulting in ORF7b truncation and the cessation of ORF8 transcription, was identified to possess enhanced replicative fitness in vitro and exhibited viral loads comparable to the WT strain [118]. During the Omicron outbreak, a study indicated that a deletion of 871 bp nucleotides in Omicron variants BA.2 (∆871 BA.2) might have been responsible for the deletion of ORF7a, ORF7b, and ORF8, which might shorten the hospital stay for patients infected with the BA.2 subvariant. Nevertheless, these deletions seem to have no significant impact on the clinical severity or the humoral and cellular immune responses [70].

Notably, the ORF10 exhibits no homology with any known proteins in other organisms, including SARS-CoV. Although a limited number of amino acid substitution mutations are observed in ORF10, which is likely attributable to ribosomal frameshifting, these alterations do not appear to compromise functionality. Upon proteomic analysis, ORF10 emerges as one of the proteins with a significant density of potential promiscuous and immunogenic cytotoxic T lymphocyte (CTL) epitopes. Specifically, ORF10 contains 11 out of 30 promiscuous epitopes, with 9 of the 11 epitopes being immunogenic CTL epitopes. Studies have independently identified reactive T cells responsive to these epitopes [119].

### 4.4. Cross-Species Transmission

The cross-species transmission of SARS-CoV-2 is an important issue that could accelerate the viral evolution and provide a source of new strain emergence [120]. The spillover of SARS-CoV-2 into new hosts may enhance the potential for recombination and evolution, consequently leading to its transmission back to humans and viral fitness improvement. It was reported that 25 animal species could be naturally infected by SARS-CoV-2: including cats, dogs, mink, otters, ferrets, lions, tigers, pumas, snow leopards, gorillas, white-tailed deer, fishing cats, binturong, coatimundi, spotted hyena, Eurasian lynx, Canada lynx, hippo, hamster, mule deer, giant anteater, West Indian manatee, black-tailed marmoset, common squirrel monkey, and big hairy armadillos [121,122,123].

With the occurrence of VOCs, the potential host spectra and cross-species transmission capability of SARS-CoV-2 might be dramatically changed [55]. It was speculated that the progenitor virus of Omicron was transmitted from humans to mice, rapidly mutated to adapt to the host, and then jumped back to humans, along with the hypothesis that BA.1 may be resulting from the continuous evolution of SARS-CoV-2 in mice [51,124]. It was suggested that certain mutations within RBD can enhance viral infectivity in cells that express ACE2 orthologues, especially the ACE2 of mice, ferrets, and horseshoe bats. For example, substitution Q493K, Q498H, and N501Y in RBD can promote the adaptation of SARS-CoV-2 in mice, while mutations Y453F, F486L, and N501T were also observed in the mink-adapted strain. A study showed that Omicron BA.1 expands its receptor-binding spectra to the rodent, palm civet, and various bat species compared with the WT and Delta. Furthermore, a study showed that the R493Q mutation in BA.2 increases the binding affinities of RBD to several animal ACE2s, including rabbits, horses, pigs, goats, and sheep. BA.4/5 harboring the R493Q substitution has an equivalent or even higher binding capacity for these species than BA.2, suggesting the BA.4/5 subvariant may have increased interspecies transmission risk from natural hosts to these domestic animals [55].

As human activities contribute to global warming and the disruption of diverse ecosystems, the buffer zone between human societies and nature is progressively diminishing. Undeniably, this will result in an increased frequency of contact between humans and wildlife, subsequently promoting the transmission of viruses across species [125]. It is within this communication that SARS-CoV-2, and other similar viruses, accumulates fitness advantages, carrying the potential to trigger the next pandemic.

## 5. Strategies for Preventing Emerging Variants

Owing to their great infectivity and immune evasion compared to other previous lineages, the XBB sublineages, such as EG.5.1, FL.1.5.1, and XBB.1.16.6, gradually became the current dominant variants. However, the emerging Omicron subvariants exhibit significant resistance to most approved mAbs and the vaccines demonstrate reduced effectiveness. Therefore, further research is necessary to explore effective prevention strategies and treatments.

In addition to non-pharmaceutical intervention (NPI) measures such as social distancing, face masks, and close contact tracing, which have been proven to significantly curtail the transmission of various viral variants [126], interventions rooted in pharmaceuticals and vaccines may prove more salient in the post-pandemic era of COVID-19. Noteworthy in the realm of therapeutic drug development is the SARS-CoV-2 main protease inhibitor, exemplified by nirmatrelvir, exhibiting a broad spectrum of therapeutic efficacy [127,128]. Nevertheless, the risk of viral mutations within the 3CL protease persists, with multiple mutations identified to exert profound impacts on the antiviral potency of nirmatrelvir [129]. Addressing this challenge through interdisciplinary approaches such as machine learning and structural biology appears to be a prudent strategy [130].

In the realm of mAbs, the formidable escape capability of variants such as the XBB sublineage has dealt a devastating blow to approved antibodies, making broad-spectrum development an inescapable challenge for a successful SARS-CoV-2 mAb. Thus far, mAbs based on non-immunodominant RBD epitopes [76] and conformational locking [77] have demonstrated broad-spectrum neutralizing characteristics against SARS-CoV-2 in the laboratory phase, offering a promising solution to the dilemma of mAbs lagging behind in updates amid the virus’s frequent mutations. Furthermore, from an alternative perspective, the development of a respiratory-administered ACE2 protein, given the high affinity of SARS-CoV-2 for ACE2, represents a rational avenue for designing broad-spectrum neutralizing drugs [131].

As one of the most efficacious measures for disease control, vaccination stands paramount in the prevention of SARS-CoV-2. The development of a broad-spectrum and highly effective vaccine poses an inescapable challenge. The prevailing development strategy currently in practical application involves the selection of prevalent viral strains as antigens, exemplified by the United States FDA-approved 2023–2024 formula designated as XBB.1.5 monovalent. However, unequivocally, the unavoidable time lag in vaccine development may lead to substantial antigenic disparities between vaccines formulated based on prevalent viral strains during development and the dominant strains in the actual application period. Currently, diverse research teams globally are indefatigably dedicating themselves to the development of broad-spectrum vaccines. Various strategies, such as mosaic nanoparticles [127], mutation patching [128], antigenic distancing [129], and targeted cellular, innate, and trained immunity [130], are continually emerging. It is believed that amidst the interdisciplinary evolution encompassing artificial intelligence, materials chemistry, structural biology, and beyond, genuinely pragmatic strategies for the development of broad-spectrum vaccines can be advanced.

Ultimately, beyond direct intervention measures, sustained monitoring of SARS-CoV-2 plays a crucial role in controlling variants. Continuous surveillance of the virus in human and other animal hosts, as well as in alternative viral reservoirs such as wastewater, will maintain a lucid comprehension of the viral evolutionary trajectory. This ongoing vigilance facilitates the early identification of variant strains that undergo substantial changes in adaptability, enabling prompt adjustments to preventive and control measures.

## 6. Conclusions

Since its discovery at the end of 2021, Omicron and its sublineages have undergone continual evolution through mutations, conferring fitness advantages through various molecular biological modifications. From the initial variants BA.1 and BA.2 to sublineages such as BA.2.75 and BA.4/5, further to the internally recombined viral strain XBB within Omicron, and presently the predominant strain EG.5.1, the Omicron lineage has manifested a discernible but internally unelucidated evolutionary trajectory. Drawing on extant research evidence, we delve into several key molecular biological features that may significantly alter viral adaptability, namely, immune escape capability, cellular infectivity, and cross-species transmission potential, exploring the impact of mutations on the adaptability of Omicron subvariants.

In the context of immune escape, commencing with the emergence of the BA.1 variant, each replacement of prevalent viral strains has posed a formidable challenge to extant mAbs, vaccines, and even sera from prior infections. The immunological pressure, shaped by widespread infections or vaccine-induced immunity, has engendered mutations in dominant epitopes on the virus’s RBD. Consequently, these mutations have led to a diminution in the neutralizing efficacy of both mAbs and polyclonal serum antibodies. Various variants iteratively arising along the timeline exhibit a notable capability to evade antibodies induced by their parental viral lineages. The fitness iterations, born out of the humoral immune escape, are particularly conspicuous across numerous subvariants within the Omicron lineage.

Beyond immune evasion, changes in the inherent infectivity of SARS-CoV-2 have profound implications for fitness advantages. As a mediator binding to the predominant receptor ACE2, the virus elevates its infectivity towards host cells by enhancing the stability of the S protein itself or its affinity with ACE2. However, distinct from inescapable evolutionary pressures of immunological barriers, alterations in ACE2 affinity by SARS-CoV-2 exhibit a more subtle transformation, manifesting a dynamically ascending trend. Regarding cell entry or tissue tropism, Omicron has undergone milestone evolution. Overall, starting from BA.1, the virus has augmented its preference for endocytosis, to some extent diminishing the selectivity for the TMPRSS2-mediated membrane fusion pathway. This alteration in virus–host interaction results in heightened affinity of the virus for upper respiratory tract tissues while concurrently reducing its affinity for lower respiratory tract tissues. This shift in characteristics may potentially steer SARS-CoV-2 towards a trajectory of non-systemic infection, laying the evolutionary groundwork for the virus to assimilate more fully into human society.

Beyond the aforementioned aspects, we also express concern regarding alterations in the virus’s cross-species transmission potential. The expansion of the host range may, on the one hand, augment avenues for human infection, and on the other hand, facilitate the accumulation of mutations or recombinations in unmonitored natural environments, thereby heightening the potential harm of the virus to humanity.

In the post-pandemic era, to address the health implications stemming from the frequent mutations of SARS-CoV-2 variants, it is imperative to intensify efforts in the development of new therapeutic drugs and vaccines. Persistent investment in broad-spectrum mAbs and vaccines is particularly crucial. Furthermore, sustained monitoring of the virus will equally contribute to fortifying preparedness for the next pandemic.

## Figures and Tables

**Figure 1 viruses-16-00184-f001:**
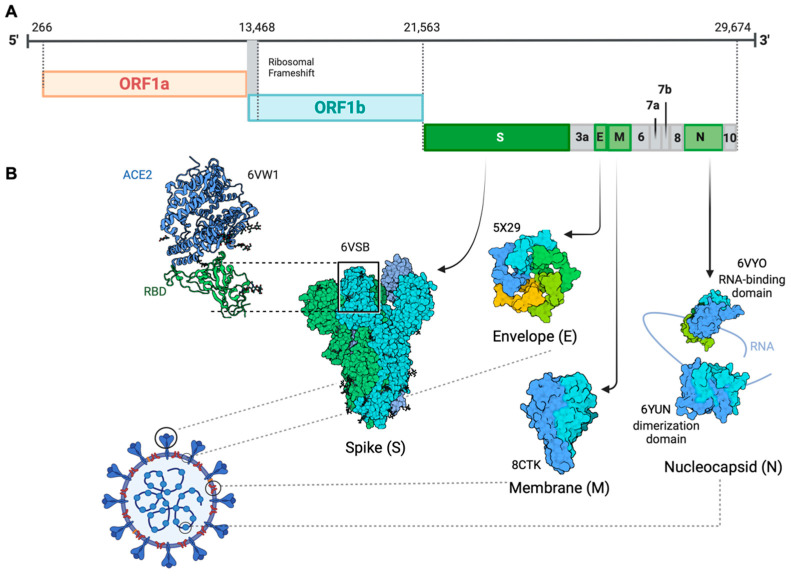
Genomic organization of SARS-CoV-2. (**A**) The 5′ end of the SARS-CoV genome consists of open reading frame (ORF) 1a/b that encodes a polyprotein. The 3′ terminus of the viral genome contains ORFs encoding the four main structural proteins, spike (S), envelope (E), membrane (M), nucleocapsid (N), ORF3a, ORF6, ORF7a/b, ORF8, and ORF10. (**B**) Protein Data Bank (PDB) identifier: spike, 6VXX; envelope, 5X29; membrane 8CTK; nucleocapsid, 6VYO and 6YUN. RBD, receptor-binding domain; ACE2, angiotensin-converting enzyme 2.

**Figure 2 viruses-16-00184-f002:**
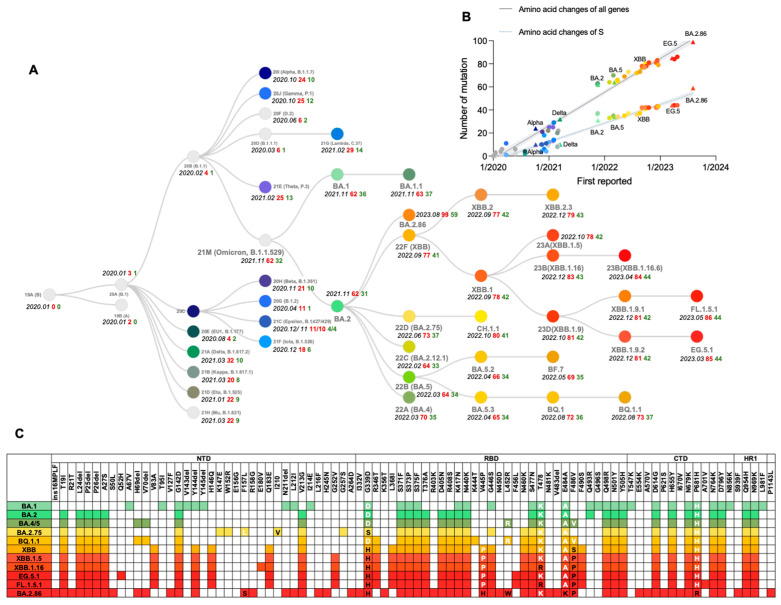
Phylogenetic relationships and the number of mutations of SARS-CoV-2 clades. (**A**) The gray font represents the names of each variant and subvariant, the black font represents the time of its initial report, the red font represents the number of all amino acid changes it has relative to wild type (18A), and the green font represents the number of S protein amino acid changes it has relative to wild type (18A). Generated from GISAID. (**B**) Scatterplots are used for linear regression evaluation of amino acid changes of all genes and S protein. (**C**) Amino acid changes of spike protein in Omicron subvariants. These diagrams were created using data from the Lineage Comparison tool from GISAID.

**Figure 3 viruses-16-00184-f003:**
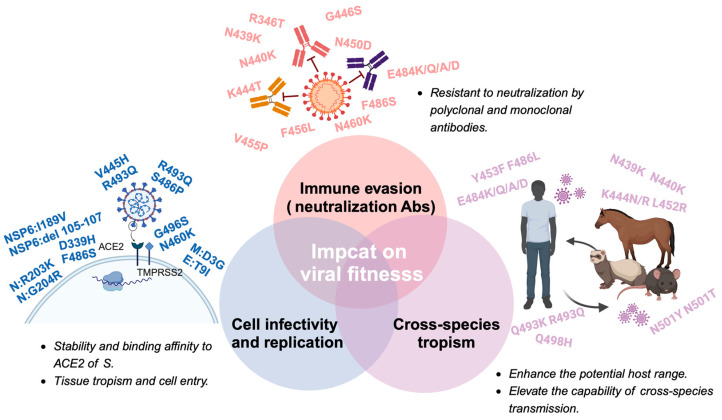
Mutations contribute to the impact on immune evasion, cell infectivity and replication, and cross-species tropism. M represents Membrane protein; N represents Nucleocapsid protein; E represents Envelope protein; NSP represents non-structural proteins; mutations without indication are all in Spike protein.

**Figure 4 viruses-16-00184-f004:**
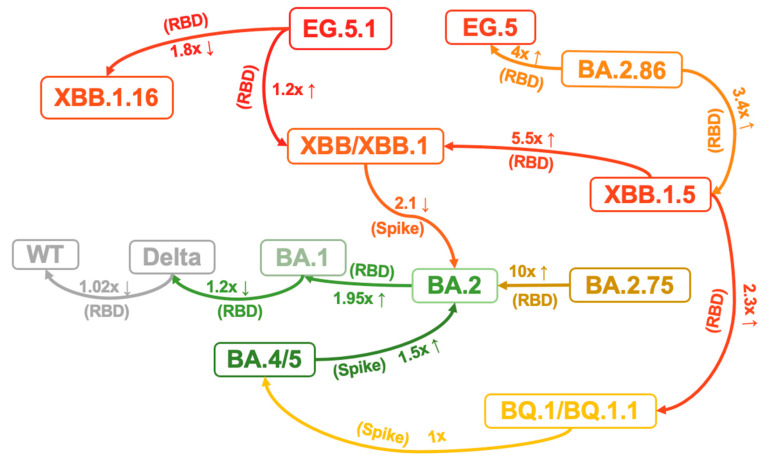
Variants net summarizes the hACE2 affinity changes of different variants in various studies [42,48,59,87,88,91]. The number represents the change in affinity relative to the variants pointed by the arrow; The superscript represents the sort order of reference: RBD characterizes the RBD from distinct variants interacting with the hACE2 via surface plasmon resonance (SPR); Spike characterizes the RBD from distinct variants interacting with the hACE2 via SPR. (It should be noted that various experimental systems are employed to assess hACE2 affinity across different studies. Although the trend is generally consistent, there lacks a feasible basis for quantitative comparison among these studies).

## Data Availability

Not applicable.

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
