# Peer review of "Evolution of the SARS-CoV-2 Omicron Variants: Genetic Impact on Viral Fitness"

_viruses, 2024, doi:10.3390/v16020184_

Round 1

Reviewer 1 Report

Comments and Suggestions for Authors

The paper of Liu W. et al entitled « Evolution of the SARS-CoV-2 Omicron variants : genetic impact on viral fitness » summarized the data on mutations, cells tropism, immune escape, mAbs and antiviral resistance, infectivity of the different Omicron variants up to the last most prevalent BA.2.86. 

This paper give a good overview of the specificity of each Omicron subvariant up to BA.2.86

I suggest the following revisions :

1/ It could interesting to add a Figure representing spike mutations of the main Omicron variants in chapter 2 « The evolution of SARS-CoV-2 » and to distinct variants evolution from variant infectivity by different subchapter.

2/ Figure 2B : please better distinct the lines representing amino acids changes between « all genes » and « S », for example with dashed line and solid line.  

In Figure 2 legend, the wildtype is term « 18A » instead of « 19A ».

3/ Line 123 : please add the Re of Delta.

4/Line 231 : please change « significant change » by « means change»

5/ Lines 323-342 : molecules Amubarvimab/Romlusevimab and Bebtelovimab are approved in China only to my knowledge and not in United-States or European Union. Please indicated these molecules are China approved.

6/Chapter 4.2.1 : the description of increase and decrase infectivity is interesting but a Figure or Table to summarize these data could help the reader.

7/ The references could be enriched for BA.2.86 with references Khan, Nature Communication, 2023 ; Willet, The Lancet, 2023 ; Nesamari, Cell Host Microbe, 2024 or for EG 5.1 with Faraone, Emerging Microbes and Infections, 2023 to enrich the data available in paragraph 4 « Impacts of mutations in Omicron on viral fitness ».    

Reviewer 2 Report

Comments and Suggestions for Authors

This is a very comprehensive, well written review of current problems of SARS-CoV-2 and Covid-19. The paper summarizes virological/biological concepts and immunological as well as clinical aspects of the disease, including vaccination strategies.

The review finds its limits when it attempts to understand,, let alone explain, the origin and high variability of the virus and in particular the emergence of the overwhelming number of omicron variants with constantly changing properties. There is a major puzzle that has not been approached in depth yet. Not in this excellent review either.
